# Radical-constructed intergrown titanosilicalite interfaces for efficient direct propene epoxidation with $H_2$ and $O_2$

Dong Lin[1,2], Xiang Feng ®[1] ✉, Yang Xu[1], Richard J. Lewis ®[2] ✉, Xiao Chen ®[3], Thomas E. Davies ®[2], Samuel Pattisson[2], Mark Douthwaite ®[2], Defu Yin[1], Qiuming He[1], Xiuhui Zheng[1], De Chen ®[4], Chaohe Yang[1] & Graham J. Hutchings ®[2] ✉

The development of titianosilicates is considered a major milestone in oxidative catalysis due to the ability of framework Ti sites to co-ordinate hydrogen peroxide/peroxy species. Herein, we demonstrate that interfacial Ti sites can be constructed through the vertical intergrowth of two MFI-type zeolite surfaces along [100] and [010] projections, with the assistance of UV-induced hydroxyl radicals. The application of these intergrown titanosilicalites as supports for Au species are observed to simultaneously offer a 2.1-fold and 3.0-fold increase in propene oxide (PO) formation rate and Au efficiency, respectively, when compared to standard Au/TS-1 catalysts. Mechanistic studies reveal that the intergrown interface Ti sites allow for lower-energy epoxidation pathways with more efficient activation of key oxygen-transfer intermediates. These results provide insights into the development of zeolite intergrown interface sites (e.g., titanosilicalite/silicalite-1/ZSM-5) and may allow for further advancements in the epoxidation of a range of key feedstocks.

Propene oxide (PO) is widely recognized as one of the most crucial chemical intermediates for the synthesis of various organic raw materials, including polyether polyols, dimethyl carbonate, and propene glycol[1–3]. It has extensive applications in the food, pharmaceutical, automotive, textile, and construction industries[4–6]. As of 2022, global demand for PO reached 11.9 million tons, and is projected to exhibit a compound annual growth rate of 5.63% until 2032[7]. Traditional chlorohydrin and hydroperoxide-based routes to propene oxide (PO) production are constrained by the generation of large amounts of byproducts, significant environmental risks, as well as the requirement for multi-step and complex processes, resulting in low atom and energy efficiencies. In light of the projected increased demand for this

major commodity chemical, the development of alternative processes with enhanced atom and energy efficiencies is essential for the chemical synthesis industry to achieve its declared sustainability targets[8,9]. The current hydrogen peroxide ($H_2O_2$) to propene oxide (HPPO) process requires the use of a stoichiometric excess of commercially produced $H_2O_2$, primarily due to the low thermal stability of the oxidant[10,11]. The direct oxidation of propane to produce propene and propene oxide has been substantiated through kinetic studies involving inert filler materials. A tentative techno-economic assessment has indicated that the direct production of propene and propene oxide from propane holds potential for practical application when compared to conventional industrial processes for propene oxide production[12].

[1]State Key Laboratory of Heavy Oil Processing, China University of Petroleum, Qingdao 266580, China. [2]Max Planck- Cardiff Centre on the Fundamentals of Heterogeneous Catalysis FUNCAT, Cardiff Catalysis Institute, School of Chemistry, Cardiff University, Cardiff CF24 4HQ, United Kingdom. [3]Beijing Key Laboratory of Green Chemical Reaction Engineering and Technology, Department of Chemical Engineering, Tsinghua University, Beijing 100084, China. [4]Department of Chemical Engineering, Norwegian University of Science and Technology, Trondheim 7491, Norway. ✉e-mail: xiangfeng@upc.edu.cn; lewisr27@cardiff.ac.uk; hutch@cardiff.ac.uk

Another promising alternative, the direct epoxidation of propene using $H_2$ and $O_2$ is gaining prominence as a future production method[13–15]. This approach is not only significantly easier to operate, featuring straightforward product separation, but also presents numerous substantial economic advantages[16–18] compared with traditional chlorohydrin and hydroperoxide-based processes.

Haruta et al. report the catalytic capabilities of Au/Ti-based catalysts for propene epoxidation using $H_2$ and $O_2$[19], have led to extensive investigations into this field with seminal contributions from a number of laboratories[20–24]. Propene epoxidation over Au/TS-1 catalysts is considered to involve a tandem mechanism operating on remote Au and Ti sites, as well as a simultaneous mechanism operating over Au and Ti species in close proximity[9,23], with the $H_2O_2$ synthesized over Au sites from the reaction between $H_2$ and $O_2$ subsequently utilized by Ti sites to facilitate propene epoxidation. In situ UV-vis and in situ X-ray absorption near edge structure (XANES) studies have led to the general acceptance that key reaction intermediates are Ti-OOH species[25,26]. Recently, an alternative perspective has emerged, suggesting that the cooperative interaction between two adjacent Ti atoms enhances the efficiency of propene epoxidation using preformed $H_2O_2$ compared to isolated Ti sites[1]. All these findings are based on two distinct active sites: Au nanoparticles and $TiO_4$ sites located in the TS-1 framework. For several decades, the isolated framework $TiO_4$ species have been considered the active sites for propene epoxidation. However, the absence of distinct synthesis methods and advanced characterization of Ti active sites in titanosilicates has hampered development in this area.

It is well-established that the hydrolysis rate of titanium precursors is typically higher than that of silicon precursors during the preparation of sol-gels[16]. This disparity leads to facile oligomerization of Ti monomers, which, consequently, results in the undesired formation of anatase $TiO_2$ phases. Several strategies have been adopted to try and suppress this, many of which include introducing additives (e.g., $H_2O_2$, isopropanol, Tween-20, $(NH_4)_2CO_3$, and Triton X-100)[16,21,27–29] in the process. These approaches can effectively restrain the hydrolysis of the Ti precursor, promoting interactions between Ti and Si species and the template, thereby reducing $TiO_2$ formation. However, the use of additives presents significant challenges in terms of controllable synthesis of titanosilicalites with custom-designed zeolite facets, especially for intergrown zeolite facets.

In this study, hydroxyl free radicals, generated in situ through ultraviolet irradiation without the use of additives, were employed to facilitate the construction of vertical intergrowth of MFI zeolite surfaces along the [100] and [010] projections. Meanwhile, hydroxyl radicals also inhibit the formation of anatase $TiO_2$. This method not only constructs intergrown zeolite surfaces in Ti-containing TS-1 but also in pure silicate-1 (S-1) and Al-containing ZSM-5. Serving as representative examples, the resulting titanosilicalites, enriched with intergrown interface sites and subsequently loaded with Au, simultaneously demonstrated a PO formation rate of approximately 318 $g_{PO}h^{-1}kg_{Cat}^{-1}$ and Au efficiency of about 460 $g_{PO}h^{-1}g_{Au}^{-1}$. Combined with in situ UV-vis, in situ FT-IR, DFT calculations, iDPC-STEM and multiple characterizations, it is found that the superior reaction performance can be attributed to vertically intergrown interface sites, which effectively promote the formation of more electrophilic oxygen-transfer intermediates. As a result, the overall energy of the reaction pathway is reduced in comparison to traditional framework sites.

## Results

### Vertical intergrowth of zeolite surfaces

It is well established that UV irradiation into water can promote the formation of hydroxyl radicals ($\cdot$OH)[16,30]. In light of this, the sol-gel precursors of TS-1 were exposed to UV radiation, at power levels ranging from 200 to 1000 W, to assess how the formation of these radical species influences the nucleation process and subsequent crystal structure.

The EPR spectra (Fig. 1a) of the titanosilicalite synthesis gel containing the radical-trapping agent (5,5-dimethylpyrroline-$N$-oxide, DMPO) under UV irradiation for 1 h with various power levels (ranging from 200 to 1000 W) reveal the nature of the radical species generated during the titanosilicate synthesis procedure. The DMPO-hydroxyl free radicals adduct showed a typical 1:2:2:1 quartet pattern with characterized hyperfine coupling constants ($a_N = a_{H\beta} = 15.0$ Gs)[30,31]. The EPR signals with $a_N = 16.0$ Gs and $a_{H\beta} = 22.0$ Gs were attributed to DMPO-carbon-centered radicals adducts[16], and the generation of carbon-centered radicals was ascribed to the interaction between $\cdot$OH and the silicon source (TEOS) during the dissociation process[32]. The signals associated with $\cdot$OH-DMPO adducts and DMPO-carbon-centered radicals adducts were both observed after UV irradiation. As the power of the UV source was increased from 200 to 1000 W, the relative intensity of the $\cdot$OH signals increased from 55 to 100%, due to increased $H_2O$ homolysis by enhanced UV irradiation[33,34] (Fig. 1b). Interestingly, the relative intensity of the carbon-centered radicals was nearly unchanged (-13%) below 300 W, but increased significantly after 500 W (74%) of power was applied. This suggests that the effective dissociation of TEOS into Si monomer ($Si(OH)_4$) for crystallization requires sufficient relative intensities (90%) of hydroxyl free radicals, higher than 500 W.

X-ray Diffraction (XRD) patterns of all titanosilicalite samples irradiated by different power levels (ranging from 200 to 1000 W) showed five peaks, which are characteristic of the MFI topological structure[35,36] (Fig. S2). The bands at 550, 800, and 1230 $cm^{-1}$ in FT-IR spectra (Fig. S3) are consistent with this and also provide evidence of an MFI topological structure. The band at 960 $cm^{-1}$ is indicative of a framework Ti atom[27,37]. The relative crystallinity was evaluated by integrating the area under the XRD spectra within the 6–40° range. TS-1–1000 W, which exhibits the highest crystallinity, was assigned a value of 100%, serving as the reference for determining the crystallinity of other samples in comparison. The relative crystallinity was found to correlate with UV irradiation (Fig. 1c), considered to be due to the more efficient dissociation of TEOS by hydroxyl free radicals. That is, a greater concentration of hydroxyl free radicals, stimulated by stronger UV, was found to improve the crystallization of TS-1 (Fig. 1d).

SEM was subsequently employed to elucidate the morphological changes of the TS-1 samples induced by the hydroxyl free radicals (Fig. 1e). As the quantity of hydroxyl free radicals increased, the surfaces of the TS-1 samples became smoother, which can be attributed to the higher crystallinity. This gradually enhanced crystallinity would further reduce titanosilicate defects, which are represented by silanol groups (Si-OH)[38]. The relative content of Si-OH ($Q_3/Q_4$)[29,39] decreased from 22.1 to 17.0% (Fig. 1f–i) which further demonstrated that the presence of hydroxyl free radicals could heal titanosilicate defects (Si-OH) and enhance crystallinity. The total acid density of different TS-1 samples (Fig. S4 and Table S1) gradually decreased from 2.67 × $10^{-2}$ to 1.58 × $10^{-2}$ mmol/g due to decreased defects (Si-OH), which is in accordance with $^{29}$Si MAS NMR, SEM, and XRD results.

The TS-1–200W sample exhibited a rough surface with some particles, which were confirmed to be $TiO_2$ nanoparticles by UV-vis analysis (signal at ~320 nm)[28,40] (Fig. S5). Hydroxyl free radicals could be used to restrain the formation of $TiO_2$ by depolymerizing Ti oligomers, as outlined in our previous studies[16], but this requires sufficient relative intensity of hydroxyl free radicals (>70%) (Fig. 1b). Furthermore, the TS-1–500 W and TS-1–1000 W materials both showed vertically intergrown morphology, which correlates well with the significantly increased relative intensity of carbon-centered radicals (>74%) in Fig. 1b. The TS-1–1000 W sample exhibited a more pronounced development of a vertically intergrown structure compared to the TS-1–500 W analog (Fig. 1e). Therefore, it is possible to conclude that the concentration of hydroxyl free radicals may effectively restrain the formation of $TiO_2$ species and dissociate Si sources to improve crystallization via the promotion of a vertically intergrown structure and decreased defects (Si-OH groups).

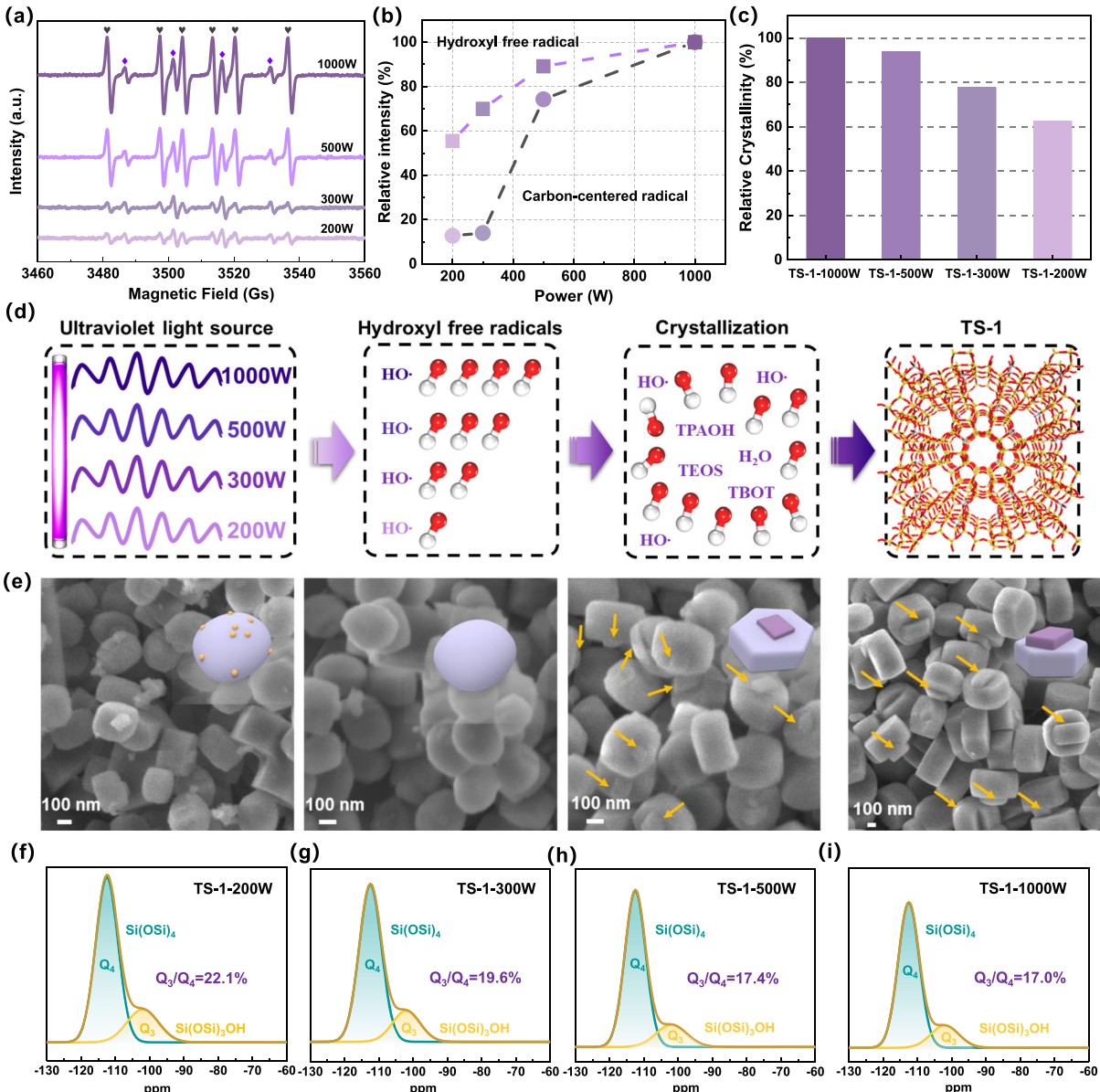

**Fig. 1 | Synthesis of TS-1 by hydroxyl radicals. a** EPR spectra of titanosilicalite synthesis gel containing the spin-trapping agent DMPO under UV irradiation for 1 h with various power levels (ranging from 200 to 1000 W, as indicated in the legend). EPR signals are labeled as follows: purple rhombus, hydroxyl free radicals; gray hearts, carbon-centered radicals. **b** The relative intensity of hydroxyl free radicals and carbon-centered radicals with various UV power levels from 200 to 1000 W.

**c** The relative crystallinity of TS-1-200W, TS-1-300W, TS-1-500W, and TS-1-1000W. **d** The schematic diagram of TS-1 crystallization under UV irradiation with various power levels. **e** Typical SEM images of TS-1-200W, TS-1-300W, TS-1-500W, and TS-1-1000W (from left to right). The intergrown surfaces on TS-1-500W and TS-1-1000W were highlighted by yellow arrows. **f–i** $^{29}$Si MAS NMR spectra of different TS-1 samples synthesized with various UV power levels.

This vertically intergrown structure was further analyzed by selected area electron diffraction (SAED), from three different angles. The representative transmission electron microscope (TEM) images of vertically intergrown TS-1 samples (from three angles) are shown in Fig. 2a–c. The marked regions (highlighted by white circles) in the vertically intergrown TS-1 were used for the selected area electron diffraction. All SAED patterns showed sharp spots without splitting, which demonstrated that the upper and lower subunits were perfectly assembled. Due to the similar lattice constants of (020) and (200) planes, Fig. 2d, e shows overlapping contrasts of (020) and (200) planes along [100] and [010] projections[41,42]. In addition, the (002) planes both appeared in upper and lower subunits along [100] and [010] projections.

In Fig. 2f, upper and lower subunits both exhibited (020) and (200) planes along the same [001] projections. Integrated differential phase contrast-scanning transmission electron microscopy (iDPC-STEM) was used to reveal the local structures of intergrown titanosilicate (Fig. 2g–i). The iDPC-STEM image extracted from the upper structural subunit shows the characteristic surface along the [010] projection, displaying orderly ten-membered rings of straight channels in Fig. 2h. Additionally, the image indicates that some Ti atoms (marked by white dots), were incorporated into the framework of the intergrown titanosilicates in Fig. 2i. Based on the structure analysis of this vertically intergrown structure, the units along [010] and [100] planes were constructed, as shown in Fig. 2j. Due to the slight mismatch between intergrown (020) and (200) planes, partially missing O

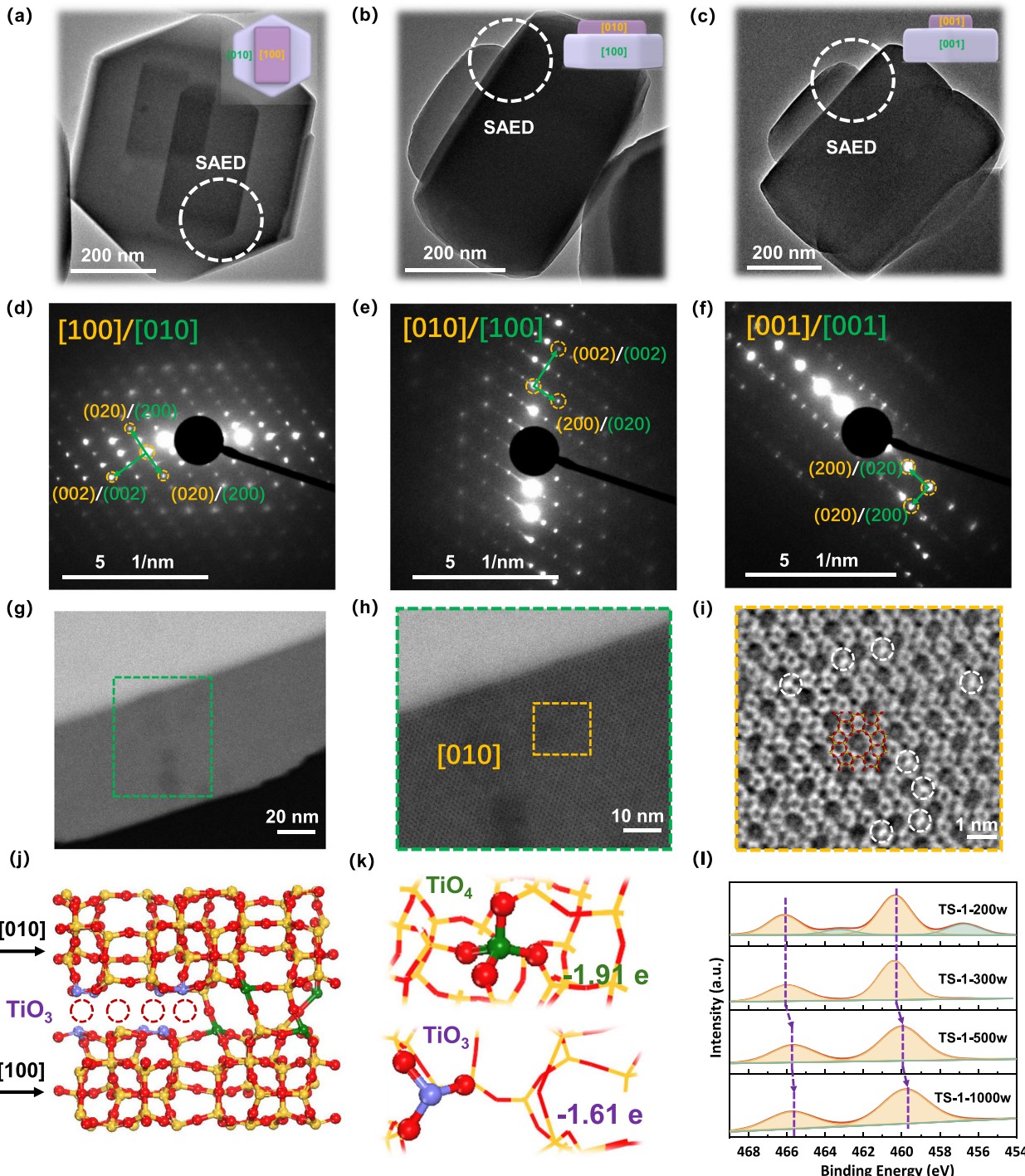

**Fig. 2 | Vertical intergrowth of zeolite surfaces.** Typical TEM images of vertically intergrown TS-1 along (**a**) [100], **b** [010], and **c** [001] projections of the upper structural subunit. The inset shows the schematic diagram of vertically intergrown TS-1 along [100], [010], and [001] projections of the upper structural subunit. The SAED patterns associated with the regions highlighted by white circles in vertically intergrown TS-1, along the (**d**) [100], **e** [010], and **f** [001] projections of the upper structural subunit. The iDPC-STEM images of intergrowth titanosilicliates at different scale bars (**g**) 20 nm, **h** 10 nm, and **i** 1 nm. **j** The interfacial model of intergrown TS-1 with partially missing O atoms due to the slight mismatch between intergrown surfaces. Red, yellow, purple and green balls represent O, Si, Ti(-O₃), and Ti(-O₄) atoms, respectively. **k** The structural model of TiO₄ and TiO₃ sites based on traditional TS-1 and vertically intergrown TS-1. The -1.91 and -1.61 represent electron loss of the Ti atom in the form of TiO₄ and TiO₃, calculated by Bader charge analysis, respectively. **l** The XPS spectra of Ti 2p over different TS-1 samples.

atoms could cause the formation of TO₃ sites (purple balls)[43]. Bader charge analysis was further used to analyse the charge transfer in TiO₄ and TiO₃ sites (Fig. 2k). The traditional TS-1 (TiO₄ sites) and vertically intergrown TS-1 (TiO₃ sites) were constructed according to the structural information above. It was found that the Ti sites of intergrown interfaces could lose 1.61 electrons while the Ti sites of traditional frameworks could lose 1.91 electrons.

The series of TS-1 samples were further characterized by XPS (X-ray photoelectron spectroscopic) analysis of the Ti 2p region, revealing the chemical properties of Ti sites (Fig. 2l). The signals at 456.8 and

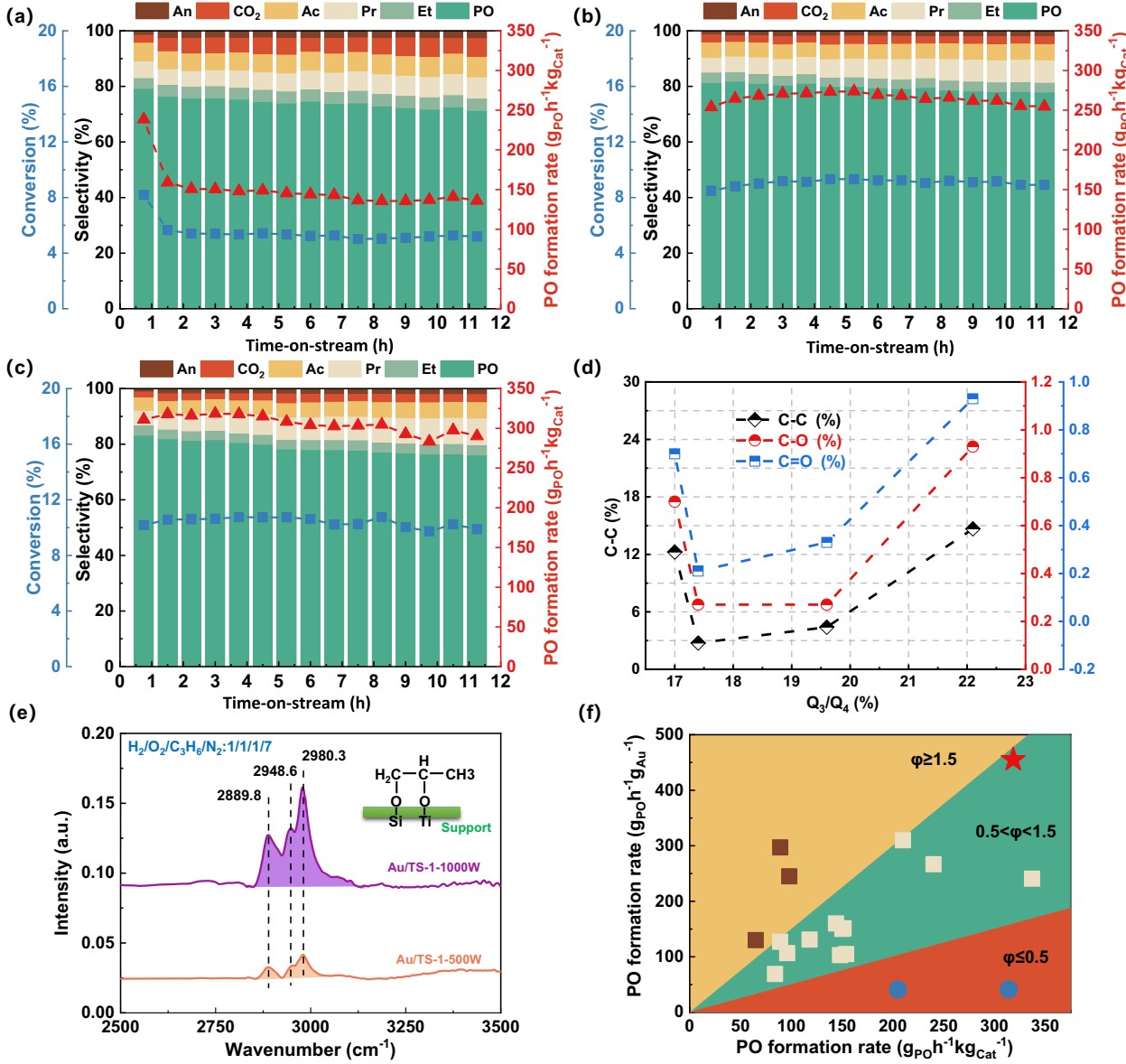

**Fig. 3 | Direct propene epoxidation with $H_2$ and $O_2$.** The PO formation rate, propene conversion and reaction selectivity of (**a**) Au/TS-1-200W, **b** Au/TS-1-300W, and **c** Au/TS-1-500W. **d** The relationship between the $Q_3/Q_4$ (Si-OH/Si-OSi) ratio and content of different carbon species (C-C, C-O. and C=O species). **e** In situ FT-IR spectroscopy results of the Au/TS-1-500W and Au/TS-1-1000W under propene epoxidation conditions (flow rate ratio of $C_3H_6/H_2/O_2/N_2$: 1/1/1/7). **f** Comparison of different reaction performances in the literature and this work (marked as red star).

Brown box: $\varphi \geq 1.5$; Apricot box: $0.5 < \varphi < 1.5$; Blue circle $\leq 0.5$. The coefficient ($\varphi$) was defined as $(g_{PO}h^{-1}g_{Au}^{-1})/(g_{PO}h^{-1}kg_{Cat}^{-1})$. Oxygenated products include propene oxide (PO), acrolein (An), propanal (Pr), acetone (Ac), ethanal (Et), and carbon dioxide ($CO_2$). Direct propene epoxidation with $H_2$ and $O_2$ conditions: reaction temperature (200 °C), catalyst (0.15 g), $H_2/O_2/C_3H_6/N_2$ = 1:1:1:7, space velocity of 14000 mL•$h^{-1}$•$g_{Cat}^{-1}$.

463.4 eV in TS-1–200 W could be attributed to $Ti^{3+}$ species from $TiO_2$[44,45], which is in agreement with observations made by UV-vis (Fig. S5) and UV-Raman (Fig. S6) spectra. Moreover, with the increase of UV power levels from 200 to 1000 W, the peaks at ~460 eV (Ti $2p_{3/2}$) and ~466 eV (Ti $2p_{1/2}$), attributed to framework Ti species, which are consistent with the Py-IR spectra in Fig. S7, gradually shift to lower binding energies. This demonstrated that the number of $TiO_3$ sites with lower valence states[46–48], gradually increased due to the formation of vertically intergrown structures, which is consistent with Bader charge results. These results demonstrated that sufficient hydroxyl free radicals could effectively construct vertically intergrown TS-1 with rich interface sites and decreased defects (Si-OH). Furthermore, these hydroxyl free radicals have the potential to construct the distinctive

vertically intergrown structure not only in Ti-containing TS-1 but also in pure siliceous S-1 and Al-containing ZSM-5 samples (Fig. S8).

## Direct propene epoxidation with $H_2$ and $O_2$

These TS-1 samples with a similar Si/Ti ratio (~200, Table S2) were loaded with Au nanoparticles (~0.07 wt%) and further tested in direct propene epoxidation with $H_2$ and $O_2$ (Fig. 3a–c and Fig. S9). High-resolution transmission electron microscopy (HRTEM) analysis of all Au/TS-1 samples depicted in Fig. S10a–d showcase a uniform size distribution among the Au nanoparticles, all exhibiting an approximate average size of 2.2 nm. Moreover, these Au/TS-1 catalysts have similar $Au^0$ percentages of ~90% (Fig. S11a–d), which are considered as intrinsic active sites for synthesizing $H_2O_2$ by $H_2$ and $O_2$[49]. The spatial

location of Au species was characterized using $V_{na}/V_{support}$ descriptor[50] shown in Table S3. The comparable $V_{na}/V_{support}$ values further affirm that all catalysts exhibit a similar spatial distribution of Au species. These results demonstrate that all the Au/TS-1 samples possess nearly identical Au sites but differ significantly in regard to the Ti sites.

The Au/TS-1–200 W catalyst exhibited a rapid deactivation from 238 to 150 $g_{PO}h^{-1}kg_{Cat}^{-1}$ during the initial 2 h of reaction due to the presence of tiny $TiO_2$. The propene conversion decreased from 8.2 to 5.2% and the selectivity of propene oxide decreased from 79.3 to 71.3% after 11 h. In contrast, as the defects gradually diminished, the fewer Si-OH sites could enhance the desorption of PO and restrain additional side reactions. The reactivity of Au/TS-1-300W and Au/TS-1–500 W remained relatively stable throughout the reaction period.

The Au/TS-1-500W catalyst, which possessed abundant intergrown interface sites exhibited a higher stable PO formation rate (~300 $g_{PO}h^{-1}kg_{Cat}^{-1}$) than the analogous Au/TS-1–300 W catalyst (~260 $g_{PO}h^{-1}kg_{Cat}^{-1}$) which had only traditional framework Ti sites. The Au/TS-1–500 W and Au/TS-1–300 W showed similar PO selectivity (~80%). With the further growth of vertical intergrown surfaces, the Au/TS-1–1000 W demonstrated a higher initial PO formation rate (322 $g_{PO}h^{-1}kg_{Cat}^{-1}$) but poorer reaction stability in Fig. S9. During initial reaction time of 3 h, the Au/TS-1–1000 W maintained a PO formation rate of ~320 $g_{PO}h^{-1}kg_{Cat}^{-1}$ with a PO selectivity of ~80%. Then the PO formation rate gradually decreased to ~260 $g_{PO}h^{-1}kg_{Cat}^{-1}$ with a PO selectivity of ~72%. The decreased reaction stability and selectivity should be attributed to the overmuch sinusoidal channels with a (100) crystal plane demonstrated by SEM in Fig. 1e, XPS in Fig. 2l and DFT calculations in Fig. 2k. These tortuous sinusoidal channels could pose more diffusion difficulties than a straight channel with a (010) crystal plane[51,52]. The TGA profiles, as shown in Fig. S12, display the weight loss of various catalysts. The weight loss occurring between 200–800 °C and below 200 °C is attributed to carbonaceous deposits and water, respectively. It is evident that the carbonaceous deposit content increases in Au/TS-1–1000 W compared to Au/TS-1–500 W (3.21 > 3.08%). Consequently, this could lead to the generation of side products and coke during the ring-opening reactions of propene oxide, aligning with the observed reaction selectivity and stability.

The XPS spectra of C1s were further used to analyze the coke species caused by ring-opening and further polymerization reactions in different Au/TS-1 samples (Fig. 3d and Fig. S13a–d). As UV power increases, the $Q_3/Q_4$ ratio (as determined by $^{29}$Si MAS NMR) of the different titanosilicalites gradually decreases. A lower $Q_3/Q_4$ ratio is indicative of higher hydrophobicity, which is considered to result in enhanced desorption of PO molecules and the suppression of carbonaceous species formation. It is found that with the decrease of $Q_3$(Si-OH)/$Q_4$(Si-O-Si) ratio, the concentration of carbon species (C-C, C-O, and C=O)[53,54] gradually decreased, as indicated by XPS analysis. The Au/TS-1-500W showed the lowest content of C-C (2.75%), C-O (0.27%), and C=O (0.21%) species. However, the Au/TS-1–1000 W exhibited higher content of C-C (12.25%), C-O (0.7%), and C=O (0.7%) species, which is in accordance with the TGA results, low reaction stability and increased presence of sinusoidal channels with high diffusion resistance. The differences in the reaction stability due to PO decomposition over Au/TS-1-500W and Au/TS-1–1000 W with rich tortuous sinusoidal channels were further tested via in situ FT-IR characterization as shown in Fig. 3e. The bands at 2889, 2948, and 2980 $cm^{-1}$ are assigned to the C-H stretching vibrations of the bidentate propoxy species, which can result from PO decomposition in the zeolite channels[21,55]. It is found that the intensity of carbon species over Au/TS-1–1000 W after 11 h is significantly stronger than that over Au/TS-1–500 W, which is in line with XPS analysis of carbon species. Therefore, the highly developed intergrown titanosilicalite interfaces with more tortuous sinusoidal channels could cause the diffusion restriction of PO and further deep oxidation of PO for lower selectivity and more coke species. During the prolonged catalytic test of propene epoxidation with $H_2$ and $O_2$

(Fig. S14, S15), the Au/TS-1–200 W and Au/TS-1–1000 W catalysts exhibited relatively rapid deactivation and low PO selectivity. In contrast, the Au/TS-1–500 W catalyst has abundant intergrown interface sites which promote efficient propene epoxidation, and high hydrophobicity which suppresses the ring-opening reactions of PO to minimize the formation of side products and coke deposits, thereby achieving a relatively stable PO formation rate and high selectivity.

The φ was defined as a ratio of the PO formation rate per gram of Au and the PO formation rate per gram of catalyst. When φ is equal to or larger than 1.5, the catalyst can be considered to exhibit a high PO formation rate per gram of catalyst but a low Au utilization efficiency. When φ is equal to or less than 0.5, the catalyst can be considered to exhibit a low PO formation rate per gram of catalyst but high Au utilization efficiency. It should be noted that the Au/TS-1-500W catalyst with rich intergrown interface sites not only offered high PO formation rate per gram of catalyst (~310 $g_{PO}h^{-1}kg_{Cat}^{-1}$), but also high PO formation rate per gram of Au (450 $g_{PO}h^{-1}g_{Au}^{-1}$) compared with other catalysts reported in literature[9,13,48,56–63] (Fig. 3f and Table S4). This Au/TS-1–500 W catalyst exhibited a 2.1-fold increase in the propene oxide (PO) formation rate and a 3.0-fold improvement in Au efficiency compared to typical Au/TS-1 catalysts of propene epoxidation (~150 $g_{PO}h^{-1}kg_{Cat}^{-1}$ and ~150 $g_{PO}h^{-1}g_{Au}^{-1}$)[13,17,21,57]. Therefore, sufficient hydroxyl free radicals irradiated by high UV power could effectively construct vertically intergrown TS-1 with rich interface sites and decreased defects (Si-OH) to enhance the PO formation rate and PO selectivity. However, overdeveloped intergrown interfaces with more tortuous sinusoidal channels, induced by the excessive generation of hydroxyl free radicals from exceptionally high UV power, could cause the diffusion restriction of PO and further deep oxidation of PO for lower selectivity and more coke species.

## Epoxidation mechanism on intergrown interfacial sites

Density functional theory (DFT) calculations were employed to examine the mechanistic aspects of propene epoxidation with $H_2$ and $O_2$ catalyzed by traditional framework and intergrown interface active sites. The interface sites were constructed through the intergrowth of crystal planes along the [010] and [100] directions, as verified through SAED and XPS characterizations. Given the inherent similarity of Au species across the catalyst series, as determined by XPS and HRTEM characterization, it is reasonable to conclude that the initial step involving the synthesis of $H_2O_2$ from $H_2$ and $O_2$ on Au nanoparticles is analogous. In addition, these different Au/TS-1 catalysts show similar $H_2O_2$ productivity (~11–13 $mol_{H2O2}kg_{cat}^{-1}h^{-1}$) between $H_2$ and $O_2$ in Fig. S16 which further demonstrated a comparable initial step of $H_2O_2$ synthesis on Au nanoparticles.

The prevailing consensus is that Au active sites play a role in $H_2O_2$ formation, while Ti active sites are involved in propene epoxidation[9,14,21]. Due to the similarity in the initial step of synthesizing $H_2O_2$ from $H_2$ and $O_2$ on Au nanoparticles, the principal focus of the DFT calculation pertains to subsequent steps involved in propene epoxidation occurring on distinct traditional framework (structure 1 in Fig. 4a) and intergrown interface (structure 1 in Fig. 4b) active sites.

In accordance with the established olefin epoxidation mechanism[1,25], the initial step involves the adsorption of $H_2O_2$ onto Ti sites. This is illustrated in structures 2 of Fig. 4a, b, corresponding to the traditional framework and intergrown interface active sites, respectively. Whether on a traditional framework or an intergrown interface active sites, the -OOH moiety is oriented inwardly toward the interior of the ten-membered ring channel. This is in accordance with the previous study[1]. Nevertheless, owing to the presence of a tricoordinated Ti atom within the intergrown interface sites, the Ti atom exhibits an inclination to coordinate with two O atoms originating from $H_2O_2$, resulting in the formation of a stable penta-coordinated adsorption structure. This bears a notable resemblance to the structure observed in the typical case of tetra-coordinated Ti atoms within

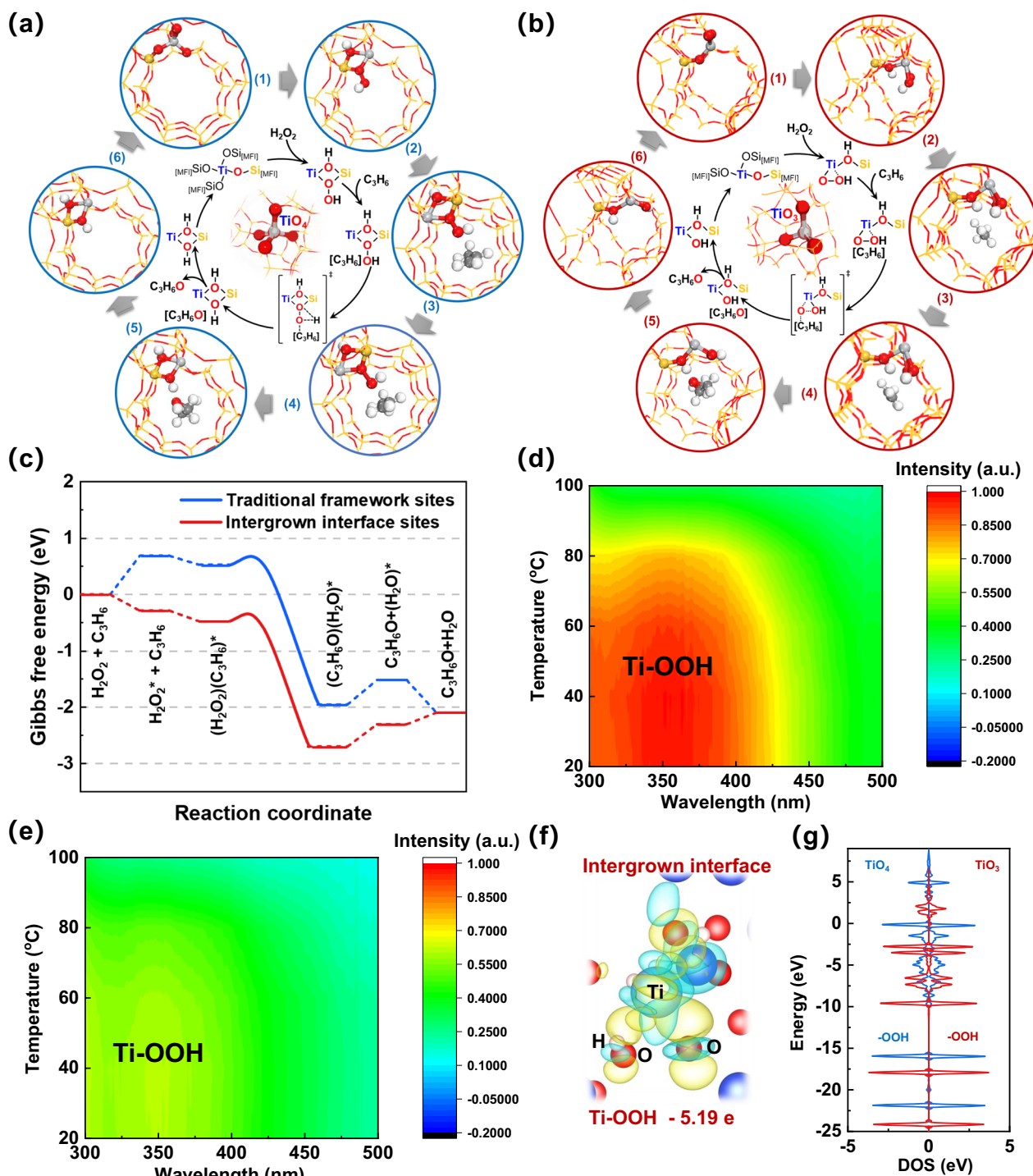

**Fig. 4 | Epoxidation mechanism on intergrown interfacial sites.** The schematic diagram of propene epoxidation with $H_2O_2$ on (**a**) traditional framework and **b** intergrown interface sites. **c** The Gibbs free energy files of propene epoxidation with $H_2O_2$ on a traditional framework and intergrown interface sites. In situ UV-vis spectra of (**d**) Au/TS-1–500 W (intergrown interface sites) and **e** Au/TS-1–300 W (traditional framework sites) after contact with performed $H_2O_2$. Experimental conditions: 0.15 g of TS-1 samples were used in conjunction with 0.01 g of $H_2O_2$, while maintaining a temperature range from 20 to 200 °C, with data points collected every 20-degree interval. **f** The differential charge density plots of $H_2O_2$ adsorbed on intergrown interface sites. Light blue and yellow isosurfaces indicate a decrease and an increase of 0.01 eV Å$^{-3}$, respectively. **g** The total density of states (DOS) of -OOH atoms on the traditional framework and intergrown interface sites.

the traditional framework sites, where the Ti atom coordinates with a single O atom from $H_2O_2$ to yield a stable penta-coordinated adsorption configuration on the Ti atom[1]. During the initial $H_2O_2$ adsorption step, the traditional framework sites gain an energy of 0.69 eV, whereas intergrown interface sites release a substantial 0.28 eV

(Fig. 4c). This discrepancy underscores the substantial enhancement of $H_2O_2$ adsorption facilitated by intergrown interface sites.

Upon the subsequent introduction of propene into the MFI channel (structure 3), the system attains further stabilization, resulting in energy release of 0.16 and 0.19 eV for the traditional framework and

intergrown interface sites, respectively. This can be attributed to non-covalent interactions that manifest between the olefin and the zeolite framework[64]. The ensuing stage involves the key transfer of oxygen from the -OOH intermediate into the propene, ultimately yielding propene oxide. The transition states, depicted in structure 4, exhibit similar activation energy barriers of 0.17 and 0.14 eV for the traditional framework and intergrown interface sites, respectively. This exothermic step in the olefin epoxidation process results in the comparable release of energy amounting to 2.48 and 2.22 eV for the traditional framework and intergrown interface sites, respectively. Following this, the propene oxide and water were progressively desorbed from channels (structure 6 and 1), facilitating the restoration of the traditional framework and intergrown interface sites. This marks the completion of the reaction cycle governing propene epoxidation. Further analysis (Fig. S17) was used to show the adsorption energy of reaction intermediates. It is found that $H_2O_2^*$ and related $H_2O^*$ species on intergrown interface sites showed significantly lower adsorption energies than those on traditional framework sites. It means that the intergrown interface sites could significantly enhance the $H_2O_2$ adsorption during propene epoxidation.

The in situ UV-vis spectra (Fig. 4d, e and Fig. S18) were further used to reveal the stronger adsorption of $H_2O_2$ on intergrown interface sites. The key oxygen-containing active intermediates, Ti-OOH, will be formed between Ti sites and $H_2O_2$. It is found that the Ti-OOH species (300–500 nm) on intergrown interface Ti sites showed a stronger signal than that on traditional framework sites (from 20–80 °C), which is in accordance with DFT calculations. After reaching 100 °C, the adsorbed $H_2O_2$ underwent a significantly increased decomposition, without evident signals remaining. Moreover, the gradual shift to higher wavelength (Fig. S18) further demonstrated more electrophilic groups on intergrown interface Ti sites[65]. The hydrogen efficiency (Fig. S19) was employed to assess the $H_2O_2$ utilization efficiency in propene epoxidation across various Au/TS-1 catalysts. Remarkably, the Au/TS-1–500 W, characterized by abundant intergrown interface sites, consistently exhibited the highest hydrogen efficiency over the 11-h reaction period. Considering these analogous Au nanoparticles with a comparable $H_2O_2$ formation rate in the presence of $H_2$ and $O_2$, the highest hydrogen efficiency of Au/TS-1-500W, with rich intergrown interface sites, demonstrated effective in situ utilization of $H_2O_2$ for subsequent epoxidation processes. These findings provide additional evidence for the efficient activation of $H_2O_2$, corroborating with DFT calculations and in situ UV-vis spectra.

The differential charge density plots presented in Fig. 4f and Fig. S20a illustrate a significant strengthening of the bond between the Ti atom of intergrown interface sites and the OOH group compared to that between the Ti atom of traditional framework sites and the OOH group. To further elucidate the electron distribution in the adsorption model, Bader charge analysis was conducted. Figure 4f reveals that Ti-OOH (intergrown interface sites) could transfer more electrons (−5.19 e) to the zeolite framework compared to traditional framework sites, thereby forming more electrophilic groups, which is consistent with in situ UV-vis results (Fig. S18). These more electrophilic groups on intergrown interface sites can enhance propene epoxidation[65].

For a deeper understanding of the electronic structure, the density of states (DOS) of Ti-OOH groups was further computed (Fig. 4g and Fig. S20b). In comparison to the density of states (DOS) of Ti and OOH groups at traditional framework sites, those at intergrown interface sites exhibit a pronounced shift toward lower-energy levels. Specifically, both the antibonding orbitals above the Fermi level and the bonding orbitals below the Fermi level for Ti and OOH groups shift downward, suggesting strengthened bonding interactions. These results underscore the role of intergrown interface sites in facilitating the formation of Ti-OOH groups and enhancing their electrophilicity.

Therefore, in comparison to traditional framework sites, intergrown interface sites exhibit a heightened capability to enhance $H_2O_2$ adsorption, consequently facilitating the formation of a greater number of electrophilic Ti-OOH groups. These electrophilic Ti-OOH groups, in turn, reduce the reaction barrier, promoting the crucial oxygen transfer from $H_2O_2$ to propene and thus establishing a more energetically favorable epoxidation pathway.

## Discussion

In summary, vertical intergrown interface sites of zeolite (e.g., TS-1, ZSM-5, and silicalite-1) surface along [100] and [010] projections are created by the presence of hydroxyl radicals. Serving as representative examples, titanosilicalites enriched with intergrown interface sites and loaded with Au, simultaneously exhibit a superior PO formation rate (~318 $g_{PO}h^{-1}kg_{Cat}^{-1}$) and excellent Au efficiency (~460 $g_{PO}h^{-1}g_{Au}^{-1}$) compared with traditional Au/TS-1 catalysts. A mechanistic investigation revealed that intergrown interface sites play a crucial role in facilitating the formation of a higher number of electrophilic Ti-OOH groups. This, in turn, reduces the epoxidation barrier, promoting the efficient oxygen transfer from $H_2O_2$ to propene and establishing a more energetically favorable epoxidation pathway. The lower-energy nature of the epoxidation pathway on intergrown interface sites, compared to conventional framework sites, is attributed to the more effective activation of key oxygen-transfer intermediates. These findings provide strategies for facet engineering of zeolites and opportunities for further optimizing related industrial epoxidation processes.

## Methods

### Synthesis of TS-1 with the assistance of ultraviolet light

A mixture was prepared by combining 17.00 g of tetra-propylammonium hydroxide solution, 21.45 g of high-purity water ($H_2O$, 18.2 MΩ•cm), 30.44 g of tetraethyl orthosilicate, and 0.25 g of tetrabutyl titanate at a stirring speed of 600 rpm. Subsequently, this mixture was exposed to ultraviolet (UV) light with varying power levels (ranging from 200 to 1000 watts, using a mercury lamp) for a duration of 1 h. The UV irradiance levels are ~0.29, 0.75, 1.12, and 2.7 mW/cm², corresponding to power levels of 200, 300, 500, and 1000 watts, respectively. Following UV irradiation, the clear mixture underwent crystallization at a temperature of 170 °C over a period of three days. The resultant synthesized material was carefully washed three times by centrifugal washing using deionized water and subsequently dried at 80 °C for 12 h. To eliminate the template, the sample was subjected to calcination at 550 °C in a static air environment, with a heating rate of 2 °C/min, lasting for 6 h. The resulting white powder was designated as TS-1-x w, with "x" denoting the power of the ultraviolet light used in the synthesis process.

### TS-1 loaded with Au particles

TS-1 loaded with Au particles were prepared via an improved impregnation method. Initially, 0.017 g of chloroauric acid was dissolved in 4.45 g of high-purity water (18.2 MΩ•cm) to form an Au solution with a concentration of $2.2 \times 10^{-3}$ $g_{Au}$/mL. Following this, a sodium carbonate solution (0.1 or 1 mol/L) was slowly added to the Au-containing solution dropwise until the pH reached the range of 7.3–7.6 at a stirring speed of 600 rpm. Subsequently, 0.5 mL of the resulting neutral and clear solution was mixed with 0.5 g of TS-1-x. The impregnated TS-1 was then dried in a vacuum oven for 16 h at 25 °C.

### Calculation of conversion, selectivity, and hydrogen efficiency

The gas reactants and products are continuously monitored using gas chromatography (Agilent 6890, GC). The detection employed flame ionization detection (FID) and thermal conductivity detection (TCD) detectors in conjunction with a Porapak Q column and a 5A column, respectively. The formulas for calculating conversion, selectivity, and

hydrogen efficiency are presented below:

Propene conversion (%):

$$= \frac{\text{(moles of carbon dioxide)}/3 + \text{(moles of acetaldehyde} \times 2)/3 + \text{moles of } C_3}{\text{moles of propene}} \times 100 \tag{1}$$

Propene oxide selectivity (%) :

$$= \frac{\text{moles of propene oxide}}{\text{(moles of carbon dioxide)}/3 + \text{(moles of acetaldehyde} \times 2)/3 + \text{moles of } C_3} \times 100 \tag{2}$$

$$\text{Hydrogen efficiency (\%) : } = \frac{\text{moles of propene oxide}}{\text{moles of } H_2 \text{ consumed}} \times 100 \tag{3}$$

$\text{Au/TS} - 1(\text{TS} - 1)\text{based PO formation rate}(g_{po}h^{-1}kg_{Cat}^{-1}) :$

$$= \frac{\text{quantity of propene oxide(g)}}{\text{time(h)} \times \text{weight of catalyst(kg)}} \tag{4}$$

$\text{Au based PO formation rate}(g_{po}h^{-1}g_{Au}^{-1}) :$

$$= \frac{\text{quantity of propene oxide(g)}}{\text{time(h)} \times \text{quantity of Au(g)}} \tag{5}$$

## Catalytic testing

The TS-1-x samples loaded with Au were evaluated in the propene epoxidation under atmospheric pressure, conducted at 200 °C. The space velocity for the direct propene epoxidation was set at 14,000 mL•h$^{-1}$•g$_{Cat}^{-1}$. A quantity of 0.15 g of catalyst (with particle sizes between 80 and 100 mesh) was introduced into a quartz tube with a thickness of 2 mm and an inner diameter of 8 mm. A blank experiment was conducted with TS-1 without Au loading in the propene epoxidation with $H_2$ and $O_2$. It was observed that there was nearly 0% conversion of propene and 0% selectivity towards propene oxide (PO).

The reactants, consisting of nitrogen (24.5 mL/min, 99.999%), oxygen (3.5 mL/min, 99.999%), hydrogen (3.5 mL/min, 99.999%), and propene (3.5 mL/min, 99.999%), were employed in the reaction. The quartz tube containing the catalysts was subjected to gradual heating, starting from room temperature and increasing at a rate of 1 K/min, until reaching a temperature of 473 K.

**$H_2O_2$ synthesis.** The autoclave was loaded with the catalyst (0.01 g) and solvent (5.6 g methanol and 2.9 g $H_2O$). The autoclave was then pressurized to 420 psi with a 5% $H_2/CO_2$ mixture, followed by the addition of 25% $O_2/CO_2$ (160 psi). The reaction mixture was cooled to 2 °C before being stirred at 1200 rpm for 30 min.

## Density functional theory calculations

The density functional theory (DFT) calculations were conducted using the Vienna ab initio simulation package (VASP). Projected augmented-wave potentials were employed to elucidate the interaction between the internal core and external valence electrons. A cutoff energy of 400 eV and the GGA-PBE exchange-correlation energy function were utilized. Plane wave basis sets were employed to expand the Kohn-Sham orbitals. Given the substantial size of the zeolite model, a $1 \times 1 \times 1$ K points grid was employed. Dispersive interactions were considered using the DFT-D3 correction approach. The density of states (DOS) for Ti and OOH groups at intergrown interface sites and traditional framework sites were calculated using the DFT + U method. The optimization was terminated when the energy difference between consecutive steps fell below $1 \times 10^{-5}$ eV, and the forces were simultaneously less than 0.03 eV/Å. According to the MFI structure, a 38 T ($Si_{38}O_{55}H_{46}Ti$) model was constructed for $TiO_4$ and a 49 T ($Si_{49}O_{68}H_{60}Ti$) intergrown model were constructed for $TiO_3$. The intergrown model was developed using the MFI crystal along both the

[010] and [100] directions. To reduce computational costs, given the model's large size, a 49 T cluster ($Si_{49}O_{68}H_{60}Ti$) was extracted from the full intergrowth structure. Terminal Si atoms were saturated with H atoms, with only the H atoms relaxed to optimize their positions. During the adsorption simulations, the Si and H atoms were fixed, while all other atoms were allowed to relax to facilitate molecular adsorption. The adsorption configurations and active sites near the Ti centers were determined through iterative optimization of the propene epoxidation reaction steps using DFT calculations.

Additionally, a Bader charge analysis, based on a rapid algorithm, was employed to quantitatively determine the charge variations of different atoms. The charge differences were visualized using the Visualization for Electronic and Structural Analysis (VESTA) software, following the equations: $\Delta\rho(r) = \rho_{total}(r) - \rho_{H2O2}(r) - \rho_{zeolite}(r)$, where $\rho_{total}(r)$ represents the total electronic density of the zeolite with adsorbed $H_2O_2$, $\rho_{H2O2}(r)$ signifies the electronic density of $H_2O_2$, and $\rho_{zeolite}(r)$ corresponds to the electronic density of the zeolite.

## Characterization

The crystal structure of the TS-1-x or Au/TS-1-x materials was examined through X-ray diffraction (XRD) employing Cu Kα radiation (X'pert PRO MPD). The relative crystallinity was assessed by calculating the area in XRD spectra within the range of 6 to 40 degrees. TS-1-1000W, demonstrating the highest crystallinity, is designated with 100% crystallinity, serving as a reference point for calculating the crystallinity values of other samples in relation to TS-1-1000W. The chemical environment of Ti was assessed via ultraviolet-visible spectroscopy (UV-vis) using a UV-2700 spectrometer. Pure barium sulfate ($BaSO_4$) was used as the background in the UV-vis spectra. The reaction intermediate of Ti-OOH at various temperatures was unveiled through in situ spectroscopy using the AvaSpec-ULS4096CL-EVO from AVANTES, shown in Fig. S1. The spectra were calibrated with the aid of the commercial diffuse reflection reference containing Teflon supplied by AVANTES. The chemical properties of Au, Ti, C, and the surface Si/Ti ratio of the Au/TS-1 were determined by X-ray photoelectron spectroscopy (XPS) using an Escalab 250Xi instrument, with the C1s binding energy (284.6 eV) serving as a reference for the charge correction. The Au loading was detected by the inductively coupled plasma optical emission spectrometry (Agilent 730, ICP-OES). The thermogravimetry analysis combined (TG, Netzsch) was used to reveal the coke content and coke species of the catalyst with a heating speed of 10 °C/min from 30 to 1000 °C. Electron paramagnetic resonance (EPR) performed on a Brookes A300 instrument was employed to detect free hydroxyl radicals in the solution. Before conducting the test, the sample underwent irradiation with varying levels of ultraviolet light (ranging from 200 to 1000 W). For the calculation of the hyperfine coupling constants ($a_N$ and $a_{H\beta}$) of hydroxyl radicals, the $a_{H\beta}$ can be determined from the equal spacing (15.0 G) between adjacent peaks, while the $a_N$ can be obtained from the center-to-center distances between the peaks, which are also 15.0 G. The bulk Si/Ti mole ratio was quantified using X-ray fluorescence (XRF) with a Panalytical Axios PW4400 instrument. The morphology and size of the TS-1 were examined through transmission electron microscopy (TEM) using a JEM-2100UHR microscope. The chemical environment of silicon (Si) in the titaniumsilicalite samples was characterized via $^{29}$Si magic angle spinning nuclear magnetic resonance (MAS NMR) spectroscopy, employing a frequency of 119.182 MHz, a pulse duration of 20 µs, a spinning rate of 15 kHz, a relaxation delay of 7 s, and data acquisition for 820 scans. The acidity and acid strength of the materials were assessed through ammonia temperature-programmed desorption ($NH_3$-TPD) using the Micromeritics Autochem 2920 II. In the standard procedure, a 100 mg sample underwent a 2-h purge with helium at 550 °C, followed by exposure to a 2% $NH_3$/He mixed gas for ammonia adsorption. Subsequently, the sample underwent a 1-h helium purge to eliminate physically adsorbed ammonia from the surface, followed by

heating to 700 °C at a rate of 10 °C/min. The acidity was further calculated according to the peak area from 80 to 300 °C based on the standard reference. The standard reference was determined using a 2% $NH_3$/He mixed gas, employing the quantitative loop of Micromeritics Autochem 2920 II. Fourier transform infrared spectroscopy (FT-IR) using an IS30 instrument was employed to capture the structural information of titanosilicalites, with KBr as a reference. Titanosilicalites samples were ground with KBr and then pressed into thin wafers, which were subsequently tested in the IS30 apparatus. Pyridine infrared spectroscopy (Py-IR) was employed to characterize the Brønsted and Lewis acid sites. The samples underwent pretreatment at 773 K for 1 h and absorbed pyridine under vacuum conditions at room temperature. The sample mass was meticulously controlled at 25 mg, and all samples were tested under identical conditions. The Nicolet iS20 FT-IR spectrometer, outfitted with an MCT-A detector, KBr windows, and a HARRICK in situ reaction chamber, was utilized to record the diffuse reflectance infrared Fourier transform spectra. A sample weighing 0.2 g was loaded into the reaction chamber. Prior to propene epoxidation with $H_2$ and $O_2$, the sample underwent in situ pretreatment with $H_2$ flowing at 25.0 ml min$^{-1}$ at 200 °C for 1 h. Background spectra were collected during continuous $N_2$ flow. Subsequently, a gas mixture of $H_2$/$O_2$/$C_3H_6$/$N_2$ (1:1:1:7, 25 ml/min) was introduced into the reaction chamber, and spectra were acquired over time until signal stabilization, about 20 min. Nitrogen was then introduced into the reaction chamber for various durations to observe changes in the catalysts.

## Data availability

All relevant data that support the findings of this study are presented in the manuscript and supplementary information file. Source data are provided with this paper. Data supporting the findings of this manuscript are also available from the corresponding author upon request. Source data are provided with this paper.

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

## Acknowledgements

This work was supported by the National Natural Science Foundation of China—Outstanding Youth Foundation (No. 22322814). D.L., R.J.L., T.E.D., S.P., M.D., and G.J.H. gratefully acknowledge Cardiff University and the Max Planck Centre for Fundamental Heterogeneous Catalysis (FUNCAT) for financial support.

## Author contributions

D.L., X.F., and R.J.L. designed the experiments. D.L., X.F., Y.X., R.J.L., X.C., T.E.D., S.P., M.D., D.Y., Q.H., X.Z., D.C., C.Y., and G.J.H. performed data analysis and experimental discussions. X.C. and T.E.D. conducted iDPC-STEM measurements and data analyses. D.L. and X.Z. carried out the DFT simulations. D.L., X.F., S.P., M.D., and R.J.L. wrote the paper. All authors discussed the results and commented on the manuscript.

## Competing interests

The authors declare no competing interests.
