## [Peer Review File · Nature Communications]

Radical-constructed Intergrown Titanosilicalite Interfaces for Efficient Direct Propene Epoxidation with H₂ and O₂

Corresponding Author: Professor Graham Hutchings

Version 0:

Reviewer comments:

Reviewer #1

(Remarks to the Author)

The manuscript titled Radical-constructed Intergrown Titanosilicalite Interfaces for Efficient Direct Propene Epoxidation with H₂ and O₂ reports very interesting and original results, where interfacial Ti sites were constructed through vertical intergrowth on titanosilicalite, silicalite-1, and ZSM-5 materials, facilitated by UV-induced hydroxyl radicals. Different UV powers, ranging from 200 W to 1000 W, were utilised to understand the radical production and catalytic properties. The catalysts were characterised using advanced in situ spectroscopic techniques and the mechanism was deduced using DFT calculations. The Au/TS-1 catalyst prepared with 500 W UV irradiation demonstrated high selectivity and production of propylene oxide in the direct epoxidation reaction.

Here are a few comments to address before recommended publication:

1. Please check the manuscript title provided in the ESI. It should be the same as the main manuscript.
2. It is recommended to either use propene or propylene throughout the manuscript for consistency.
3. Please elaborate what is meant by offer improved atom and energy efficiencies in line 35.
4. In line 102, explain how a_N and $a_{H\beta}$ is calculated for the EPR spectra shown in Figure 1a.
5. It is slightly hard to distinguish the colours of spectra in Figure 1a, especially for 300W and 500W.
6. In line 131, the authors have reported the relative crystallinity shown in Figure 1c. For easy understanding, it is recommended to refer to the calculation of relative crystallinity within the main text as explained in line 510.
7. In the line 256-257, it is mentioned that "The decreased reaction stability and selectivity should be attributed to the overmuch sinusoidal channels with a (100) crystal plane demonstrated by SEM, XPS and DFT calculations". Please refer to the figure numbers of SEM, XPS and DFT.
8. In line 268, describe how Q3/Q4 is calculated from the XPS spectra? Wouldn't this be from Si MAS NMR?
9. The authors have tested the catalyst for 12 h. It is advised to perform the test for longer time-on-stream to understand the catalyst stability, as it is an important parameter to estimate catalyst performance.
10. Were the catalysts analysed after the reaction using TEM, was there any sintering of gold nanoparticles observed? This is also one of the main reasons for catalyst deactivation. (Chemical reviews, 107(6), pp.2709-2724; Angewandte Chemie, 133(33), pp.18333-18341; The Journal of Physical Chemistry B, 109(41), pp.19309-19319.)
11. Py-IR spectra in Figure S7 are not referred to in the main manuscript (p.26?).
12. The Au/TS-1-200W catalyst exhibited a rapid deactivation, which is attributed to the presence of tiny TiO₂. The TGA profiles shown in Figure S12 illustrates that Au/TS-1-200W exhibits ~3.61% weight loss, which is the highest amongst all the catalyst. Can the authors comment on that? The deposition of carbonaceous species also had an impact on the activity. This could be further verified from in-situ FTIR. Also, it will be useful to add the TGA of fresh catalyst for comparison.
13. The article is generally very well written. Minor revisions on p.3: "Haruta et al. report" instead of "reports into"; "proximity, With the" -> "proximity, with the". p.4: "in-situ" -> "in situ" (possibly in italics; several times); p.11: "nearly-identical" -> "nearly identical".

Reviewer #2

(Remarks to the Author)

Reviewer #3

(Remarks to the Author)

In this work the authors present a nice application of a catalyst based on titanosilicates for the epoxidation of propene to propane oxide, by means of advanced experimental techniques and quantum chemical simulations.

Given my expertise I can comment the theory part mainly. I carefully read the experimental part which is very clear and sounds. However, the theory part needs to be revised as there are missing aspects that makes the results of the simulation hard to be commented.

1) The authors used a standard GGA-PBE functional, which is certainly a suitable choice when one aims at performing a large number of calculations. However, it fails in giving estimates on the electronic structure of semiconductors and insulators, as well as it tends to overbind reaction intermediates. I suggest refining the electronic structure and performing at least single-point calculations on top of PBE optimized structures.

2) Another missing aspect is the presence (if any) of dispersions in the computational setup.

3) It is unclear if the authors refer to DFT reaction energies or Gibbs free energies when discussing energy profiles. As adsorption of gas-phase species is considered, the authors should work with free energies.

4) Some statements should be reshaped, such as "The transition states depicted in structure 4, exhibit different activation energy barriers of 0.17 eV and 0.10 eV". According to the accuracy of DFT, a 0.1 eV difference is within the error of the method, the barriers are similar. The sentence at lines 316-318 is a repetition of lines 309-311.

5) It is unclear if and how the authors modeled the (010)/(100). If so, the working mismatch should be reported and the presence of spurious effects induced by the strain should be checked.

Version 1:

Reviewer comments:

Reviewer #1

(Remarks to the Author)

I am satisfied with revisions and recommend the article for publication.

Reviewer #2

(Remarks to the Author)

Reviewer #3

(Remarks to the Author)

The authors addressed the raised points.

Reviewer Response: Radical-constructed Intergrown Titanosilicalite Interfaces for Efficient Direct Propene Epoxidation with H₂ and O₂

The authors thank all referees for their time in consideration of this manuscript, we are very grateful to receive your comments, suggestions, and queries, in addressing these we consider the quality of the manuscript has been considerably improved. In accordance with the comments received we now submit our amended manuscript. We have addressed each of the comments raised by the reviewers (listed below in black), with our responses listed in blue. Modifications to the manuscript have been highlighted in red to assist the reviewers in their assessment. We trust that the modified document is now considered to be acceptable for publication.

Reviewer #1:

The manuscript titled Radical-constructed Intergrown Titanosilicalite Interfaces for Efficient Direct Propene Epoxidation with H₂ and O₂ reports very interesting and original results, where interfacial Ti sites were constructed through vertical intergrowth on titanosilicalite, silicalite-1, and ZSM-5 materials, facilitated by UV-induced hydroxyl radicals. Different UV powers, ranging from 200 W to 1000 W, were utilised to understand the radical production and catalytic properties. The catalysts were characterised using advanced in situ spectroscopic techniques and the mechanism was deduced using DFT calculations. The Au/TS-1 catalyst prepared with 500 W UV irradiation demonstrated high selectivity and production of propylene oxide in the direct epoxidation reaction.

Response:

The authors thank the reviewer for their consideration of the manuscript. We are pleased they consider the work to contain interesting results, and are grateful to receive their comments and queries, we consider that in addressing these the quality of the resubmitted manuscript has been significantly improved.

1. Please check the manuscript title provided in the ESI. It should be the same as the main manuscript.

Response:

We thank the reviewer for highlighting this, the manuscript title provided in the ESI has been revised to “Radical-constructed Intergrown Titanosilicalite Interfaces for Efficient Direct Propene Epoxidation with H₂ and O₂”.

2. It is recommended to either use propene or propylene throughout the manuscript for consistency.

Response:

According to the reviewer’s suggestion, all instances of the word “propylene” have been replaced with “propene”.

3. Please elaborate what is meant by offer improved atom and energy efficiencies in line 35.

Response:

The traditional chlorohydrin process relies on highly toxic chlorine reagents, resulting in equipment corrosion and the formation of hazardous byproducts. Moreover, the hydroperoxide-based process for propene epoxidation is hindered by its long process, high capital investment, and stringent requirements for raw material quality. Hence, the production of large amounts of byproducts, coupled with a long and complex process, leads to low atom and energy efficiencies. Therefore, enhancing atom and energy efficiencies is essential for advancing the chemical synthesis of propene epoxidation. In order to help readers better understand our results, we added and modified the following description: “Traditional chlorohydrin and hydroperoxide-based routes to propene oxide (PO) production are constrained by the generation of large amounts of byproducts, significant environmental risks, as well as the requirement for multi-step and complex processes, resulting in low atom and energy efficiencies. In light of the projected increased demand for this major commodity chemical, the development of

alternative processes with enhanced atom and energy efficiencies is essential for the chemical synthesis industry to achieve its declared sustainability targets. ”

4. In line 102, explain how a_N and $a_{H\beta}$ is calculated for the EPR spectra shown in Figure 1a.

Response:

According to the reviewer’s suggestion, the calculation of hyperfine coupling constants ($a_N = a_{H\beta}$) was given in the experimental section: “For the calculation of the hyperfine coupling constants (a_N and $a_{H\beta}$) of hydroxyl radicals, the $a_{H\beta}$ can be determined from the equal spacing (15.0 G) between adjacent peaks, while the a_N can be obtained from the center-to-center distances between the peaks, which are also 15.0 G.”

5. It is slightly hard to distinguish the colours of spectra in Figure 1a, especially for 300W and 500W.

Response:

To improve the clarity of our results, the colors of the spectra in Figure 1a have been adjusted. Additionally, labels indicating different UV power levels (e.g., 300 W and 500 W) have been added to further aid reader comprehension.

Figure 1. Synthesis of TS-1 by hydroxyl radicals. (a) EPR spectra of titanosilicalite synthesis gel containing the spin-trapping agent DMPO under UV irradiation for 1 h with various power levels (ranging from 200 to 1000 W, as indicated in the legend).

6. In line 131, the authors have reported the relative crystallinity shown in Figure 1c.

For easy understanding, it is recommended to refer to the calculation of relative crystallinity within the main text as explained in line 510.

Response:

According to reviewer's suggestion, the related sentences were added as follows: "The relative crystallinity was evaluated by integrating the area under the XRD spectra within the 6-40° range. TS-1-1000W, which exhibits the highest crystallinity, was assigned a value of 100%, serving as the reference for determining the crystallinity of other samples in comparison."

7. In the line 256-257, it is mentioned that "The decreased reaction stability and selectivity should be attributed to the overmuch sinusoidal channels with a (100) crystal plane demonstrated by SEM, XPS and DFT calculations". Please refer to the figure numbers of SEM, XPS and DFT.

Response:

According to reviewer's suggestion, the figure numbers were added as follows: "The decreased reaction stability and selectivity should be attributed to the overmuch sinusoidal channels with a (100) crystal plane demonstrated by SEM in Figure 1e, XPS in Figure 2l and DFT calculations in Figure 2k."

8. In line 268, describe how Q₃/Q₄ is calculated from the XPS spectra? Wouldn't this be from Si MAS NMR?

Response:

The ratio of Q₃(Si-OH)/Q₄(Si-O-Si) is determined from ²⁹Si MAS NMR, while the content of different carbon species (C-C, C-O and C=O) is obtained from XPS spectra. With the increase of UV power, the ratio of Q₃/Q₄ of different titanosilicalites gradually decreases. A lower Q₃/Q₄ ratio indicates higher hydrophobicity, which effectively promotes the desorption of PO molecules and inhibits the formation of carbonaceous species. This trend is consistent with the observed content of different carbon species (C-C, C-O, and C=O).

In order to help readers better understand our results, we added and modified the

following description: “As UV power increases, the Q_3/Q_4 ratio (as determined by ^{29}Si MAS NMR) of the different titanosilicalites gradually decreases. A lower Q_3/Q_4 ratio is indicative of higher hydrophobicity, which is considered to result in enhanced desorption of PO molecules and the suppression of carbonaceous species formation. It is found that with the decrease of $Q_3(\text{Si-OH})/Q_4(\text{Si-O-Si})$ ratio, the concentration of carbon species (C-C, C-O and C=O)^{53, 54} gradually decreased, as indicated by XPS analysis.”

9. The authors have tested the catalyst for 12 h. It is advised to perform the test for longer time-on-stream to understand the catalyst stability, as it is an important parameter to estimate catalyst performance.

Response:

According to the reviewer's suggestion, a longer time-on-stream was included in Figure S14 to provide a more comprehensive assessment of the catalyst's stability.

In order to help readers better understand our results, we added and modified the following description: “During the prolonged catalytic test of propene epoxidation with H_2 and O_2 (Figure S14), the Au/TS-1-200W and Au/TS-1-1000W catalysts exhibited relatively rapid deactivation and low PO selectivity. In contrast, the Au/TS-1-500W catalyst has abundant intergrown interface sites which promote efficient propene epoxidation, and high hydrophobicity which suppresses the ring-opening reactions of PO to minimize the formation of side products and coke deposits, thereby achieving a relatively stable PO formation rate and high selectivity.”

Figure S14 The (a) PO formation rate and (b) reaction selectivity of different Au/TS-1 catalysts. Direct propene epoxidation with H₂ and O₂ conditions: reaction temperature (200°C), catalyst (0.15 g), H₂/O₂/C₃H₆/N₂=1:1:1:7, space velocity of 14000 mL·h⁻¹·g_{cat}⁻¹.

10. Were the catalysts analysed after the reaction using TEM, was there any sintering of gold nanoparticles observed? This is also one of the main reasons for catalyst deactivation. (Chemical reviews, 107(6), pp.2709-2724; Angewandte Chemie, 133(33), pp.18333-18341; The Journal of Physical Chemistry B, 109(41), pp.19309-19319.)

Response:

The different Au/TS-1 catalysts were analyzed after propene epoxidation after 20 h. It was found that all Au/TS-1 catalysts exhibited similar Au particle sizes (~2.4 nm) before and after the reaction in Figure S15. We direct the reviewer to the figure below reporting particle size of the various used samples. This observation is consistent with previous studies, further confirming that carbonaceous deposition is the primary cause of deactivation in propene epoxidation with H₂ and O₂ (*ACS Catal.*, 2018, 8(11): 10649-10657; *ACS Catal.*, 2017, 7(4): 2668-2675; *Appl. Catal. B: Environ.*, 2014, 150: 396-401; *Chem. Eng. J.*, 2019, 377: 119954).

In order to help readers better understand our results, we added the following description: “Post-reaction HRTEM analysis of the different Au/TS-1 catalysts revealed that their Au particle sizes (~2.4 nm) remained nearly unchanged before and after propene epoxidation (Figure S15).”

Figure S15 Typical STEM images and associated particle count histograms of (a) Au/TS-1-200W, (b) Au/TS-1-300W, (c) Au/TS-1-500W and (d) Au/TS-1-1000W after reaction. Direct propene epoxidation with H_2 and O_2 conditions: reaction temperature (200°C), catalyst (0.15 g), $H_2/O_2/C_3H_6/N_2=1:1:1:7$, space velocity of $14000 \text{ mL}\cdot\text{h}^{-1}\cdot\text{g}_{\text{Cat}}^{-1}$.

11. Py-IR spectra in Figure S7 are not referred to in the main manuscript (p.26?).

Response:

We thank the reviewer for highlighting this.

According to reviewer's suggestion, we modified the following description: "Moreover, with the increase of UV power levels from 200 to 1000 W, the peaks at ~460 eV (Ti 2p_{3/2}) and ~466 eV (Ti 2p_{1/2}), attributed to framework Ti species which are consistent with the Py-IR spectra in Figure S7, gradually shift to lower binding energies."

12. The Au/TS-1-200W catalyst exhibited a rapid deactivation, which is attributed to the presence of tiny TiO₂. The TGA profiles shown in Figure S12 illustrates that Au/TS-1-200W exhibits ~3.61% weight loss, which is the highest amongst all the catalyst. Can the authors comment on that?

Response:

According to reviewer's suggestion, we added the following description: "The relative content of Si-OH (Q₃/Q₄) decreased from 22.1% to 17.0% (Figure 1f-1i) with the increase of power levels (ranging from 200 to 1000 W). TS-1-200W exhibited the highest Q₃/Q₄ ratio (22.1%), and a higher Q₃/Q₄ ratio indicates lower hydrophobicity, which is considered to hinder the desorption of propylene oxide (PO) molecules and promote the formation of carbonaceous species. As a result, Au/TS-1-200W demonstrated the highest weight loss (~3.61%) after propene epoxidation with H₂ and O₂."

The deposition of carbonaceous species also had an impact on the activity. This could be further verified from in-situ FTIR. Also, it will be useful to add the TGA of fresh catalyst for comparison.

Response:

According to the reviewer's suggestion, we have added the TGA results of fresh catalysts in Figure S12a. As shown in Figure S12a, all fresh catalysts exhibit neglectable weight loss (<0.5%).

Figure S12 The TGA profiles of different (a) fresh catalysts.

13. The article is generally very well written. Minor revisions on p.3: “Haruta et al. report” instead of “reports into” ; “proximity, With the” -> “proximity, with the”. p.4: “in-situ” -> “in situ” (possibly in italics; several times); p.11: “nearly-identical” -> “nearly identical” .

Response:

According to reviewer’s suggestion, we modified the following description: “Haruta *et al.* report”, “proximity, with the”, “*in situ*”, “nearly identical”.

Again, we thank the reviewer for the time and patience in evaluating our manuscript.

Reviewer #2:

I co-reviewed this manuscript with one of the reviewers who provided the listed reports. This is part of the Nature Communications initiative to facilitate training in peer review and to provide appropriate recognition for Early Career Researchers who co-review manuscripts

Response:

The authors thank the reviewer for their time and effort in reviewing our manuscript.

Reviewer #3:

In this work the authors present a nice application of a catalyst based on titanosilicates for the epoxidation of propene to propane oxide, by means of advanced experimental techniques and quantum chemical simulations.

Given my expertise I can comment the theory part mainly. I carefully read the experimental part which is very clear and sounds. However, the theory part needs to be revised as there are missing aspects that makes the results of the simulation hard to be commented.

Response:

The authors appreciate the reviewer's thorough evaluation of the manuscript. We are grateful for their valuable comments and questions, which have guided us in making substantial improvements to the quality of the resubmitted manuscript.

1. The authors used a standards GGA-PBE functional, which is certain a suitable choice when one aims at performing a large number of calculations. However, it fails in giving estimates on the electronic structure of semiconductors and insulators, as well as it tends to overbind reaction intermediates. I suggest refining the electronic structure and performing at least single-point calculations on top of PBE optimized structures.

Response:

Based on the reviewer's suggestion, the density of states (DOS) was recalculated using the DFT+U method based on the PBE optimized structures. The recalculated DOS for Ti and OOH groups at intergrown interface sites and traditional framework sites are presented in Figure 4g and Figure S18b. Compared to the DOS of Ti and OOH groups at traditional framework sites, those at intergrown interface sites exhibit a notable shift toward lower energy levels. Both the antibonding orbitals above the Fermi level and the bonding orbitals below the Fermi level for Ti and OOH groups shift to lower energies, indicating enhanced bonding interactions. These findings highlight the ability of intergrown interface sites to promote the formation and increase the electrophilicity of Ti-OOH groups.

Moreover, the overbinding of reaction intermediates typically observed with standard GGA-PBE was mitigated by incorporating a dispersion correction to account for van der Waals (vdW) interactions. This approach adds additional energy contributions based on pairwise atomic interactions and distance-dependent damping functions. Implementing the DFT-D3 correction improves the accuracy of DFT in predicting binding energies, molecular geometries, and adsorption behaviors in epoxidation systems where dispersive forces play a significant role (*Nature*, 2020, 586(7831): 708-713).

In order to help readers better understand our results, we added the following description: “In comparison to the density of states (DOS) of Ti and OOH groups at traditional framework sites, those at intergrown interface sites exhibit a pronounced shift toward lower energy levels. Specifically, both the antibonding orbitals above the Fermi level and the bonding orbitals below the Fermi level for Ti and OOH groups shift downward, suggesting strengthened bonding interactions. These results underscore the role of intergrown interface sites in facilitating the formation of Ti-OOH groups and enhancing their electrophilicity.” “Dispersive interactions were considered using the DFT-D3 correction approach. The density of states (DOS) for Ti and OOH groups at intergrown interface sites and traditional framework sites were calculated using the DFT+U method.”

Figure 4. (g) The total density of states (DOS) of -OOH atoms on traditional framework and intergrown interface sites.

Figure S18. (b) The total density of states (DOS) of Ti- atoms on traditional framework and intergrown interface sites.

2. Another missing aspect is the presence (if any) of dispersions in the computational setup.

Response:

Dispersive interactions were taken into account using the DFT-D3 correction method. In order to help readers better understand our results, we added the following description: “Dispersive interactions were considered using the DFT-D3 correction approach.”

3. It is unclear if the authors refer to DFT reaction energies or Gibbs free energies when discussing energy profiles. As adsorption of gas-phase species is considered, the authors should work with free energies.

Response:

Based on the reviewer's suggestion, Gibbs free energies were utilized in the energy files and applied to calculate the free energies of adsorption for various intermediates.

Figure 4. (c) The Gibbs free energy files of propene epoxidation with H_2O_2 on traditional framework and intergrown interface sites.

Figure S15 Calculated free energies of adsorption for different intermediates on TiO_4 and TiO_3 sites. Moreover, the related descriptions were modified based on Gibbs free energies as follows: “During the initial H_2O_2 adsorption step, the traditional framework sites gain an energy of 0.69 eV, whereas intergrown interface sites release a substantial 0.28 eV (Figure 4c).” “Upon the subsequent introduction of propene into the MFI channel (structure 3), the system attains further stabilization, resulting in energy release of 0.16 eV and 0.19 eV for the traditional framework and intergrown interface sites, respectively.” “This exothermic step in the olefin epoxidation process results in the comparable release of energy amounting to 2.48 eV and 2.22 eV for the traditional framework and intergrown interface sites, respectively.”

4. Some statements should be reshaped, such as “The transition states depicted in structure 4, exhibit different activation energy barriers of 0.17 eV and 0.10 eV”. According to the accuracy of DFT, a 0.1 eV difference is within the error of the method, the barriers are similar. The sentence at lines 316-318 is a repetition of lines 309-311.

Response:

According to reviewer’s suggestion, we added and modified the following description: “The transition states, depicted in structure 4, exhibit similar activation energy barriers of 0.17 eV and 0.14 eV for the traditional framework and intergrown interface sites, respectively.” “Due to the similarity in the initial step of synthesizing H₂O₂ from H₂ and O₂ on Au nanoparticles”.

5. It is unclear if and how the authors modeled the (010)/(100). If so, the working mismatch should be reported and the presence of spurious effects induced by the strain should be checked.

Response:

We appreciate the reviewer's reminder.

The intergrown model was constructed using the MFI crystal along both the [010] and [100] directions. To reduce computational costs given the large size of the model, a 49T cluster (Si₄₉O₆₈H₆₀Ti) was extracted from the full intergrowth structure. Terminal Si atoms were coordinated with H atoms, and only the H atoms were relaxed to optimize their positions. During the adsorption simulations, the Si and H atoms were fixed, while all other atoms were allowed to relax to facilitate the adsorption of molecules. The adsorption configurations and active sites near Ti centers were identified through iterative optimization of propene epoxidation reaction steps using density functional theory (DFT) calculations.

The MFI crystal along the [010] direction has an *a*-axis length of approximately 20.0 Å and a *c*-axis length of ~13.4 Å, whereas the crystal along the [100] direction exhibits a *b*-axis of ~20.0 Å and a *c*-axis of ~13.4 Å. The nearly identical crystal planes along the [010] and [100] directions allow for a seamless match, enabling crystal growth

along both directions. The intergrowth interface maintains dimensions of ~ 20.0 Å and ~ 13.4 Å, perfectly aligning with the pre-intergrowth planes to minimize strain. This strain-free interface was further confirmed by geometric phase analysis (GPA) strain maps (*Nat. Commun.*, 2022, 13(1): 2924), which indicated no significant strain at the interface along the [010] and [100] directions.

In order to help readers better understand our results, we added and modified the following description: “The intergrown model was developed using the MFI crystal along both the [010] and [100] directions. To reduce computational costs given the model's large size, a 49T cluster ($\text{Si}_{49}\text{O}_{68}\text{H}_{60}\text{Ti}$) was extracted from the full intergrowth structure. Terminal Si atoms were saturated with H atoms, with only the H atoms relaxed to optimize their positions. During the adsorption simulations, the Si and H atoms were fixed, while all other atoms were allowed to relax to facilitate molecular adsorption. The adsorption configurations and active sites near the Ti centers were determined through iterative optimization of the propene epoxidation reaction steps using DFT calculations.”

Again, we thank the reviewer for the time and patience in reviewing our manuscript. We have tried our best to address the comments from the reviewer and consider that in doing so the manuscript quality has been improved.

Response to reviewers and editorial comments: Radical-constructed Intergrown Titanosilicalite Interfaces for Efficient Direct Propene Epoxidation with H₂ and O₂

The authors thank all referees for their time in consideration of this manuscript, we are very grateful to receive your comments, suggestions, and queries, in addressing these we consider the quality of the manuscript has been considerably improved.

We note that no further comments/corrections were required and so no changes have been made.

All the requests made for editorial changes have been made as requested.